# Higher Epoxyeicosatrienoic Acids in Cardiomyocytes-Specific CYP2J2 Transgenic Mice Are Associated with Improved Myocardial Remodeling

**DOI:** 10.3390/biomedicines8060144

**Published:** 2020-05-30

**Authors:** Theresa Aliwarga, Xiaoyun Guo, Eric A. Evangelista, Rozenn N. Lemaitre, Nona Sotoodehnia, Sina A. Gharib, Darryl C. Zeldin, Qinghang Liu, Rheem A. Totah

**Affiliations:** 1Department of Medicinal Chemistry, University of Washington, Box 357610, Seattle, WA 98195, USA; tessa629@uw.edu (T.A.); eaevang@uw.edu (E.A.E.); 2Department of Physiology and Biophysics, University of Washington, Box 357290, Seattle, WA 98195, USA; xguo@uw.edu; 3Cardiovascular Health Research Unit, Department of Medicine, University of Washington, 1730 Minor Ave, Suite 1360, Seattle, WA 98101, USA; rozenl@uw.edu (R.N.L.); nsotoo@uw.edu (N.S.); 4Division of Cardiology, University of Washington, Box 356422, Seattle, WA 98195, USA; 5Computational Medicinal Core, Center for Lung Biology, Division of Pulmonary and Critical Care Medicine, Department of Medicine, University of Washington, S376-815 Mercer, Box 385052, Seattle, WA 98109, USA; sagharib@uw.edu; 6National Institute of Environmental Health Sciences, A214 Rall Building, 111 T W Alexander Dr, Research Triangle Park, NC 27709, USA; zeldin@niehs.nih.gov

**Keywords:** epoxyeicosatrienoic acids, myocardial remodeling, CYP2J2

## Abstract

Elevated *cis-*epoxyeicosatrienoic acids (EETs) are known to be cardioprotective during ischemia-reperfusion injury in cardiomyocyte-specific overexpressing cytochrome P450 2J2 (CYP2J2) transgenic (Tr) mice. Using the same Tr mice, we measured changes in cardiac and erythrocyte membranes EETs following myocardial infarction (MI) to determine if they can serve as reporters for cardiac events. Cardiac function was also assessed in Tr vs. wild-type (WT) mice in correlation with EET changes two weeks following MI. Tr mice (N = 25, 16 female, nine male) had significantly higher cardiac *cis-* and *trans-*EETs compared to their WT counterparts (N = 25, 18 female, seven male). Total cardiac *cis-*EETs in Tr mice were positively correlated with total *cis-*EETs in erythrocyte membrane, but there was no correlation with *trans-*EETs or in WT mice. Following MI, *cis-* and *trans-*EETs were elevated in the erythrocyte membrane and cardiac tissue in Tr mice, accounting for the improved cardiac outcomes observed. Tr mice showed significantly better myocardial remodeling following MI, evidenced by higher % fractional shortening, smaller infarct size, lower reactive oxygen species (ROS) formation, reduced fibrosis and apoptosis, and lower pulmonary edema. A positive correlation between total cardiac *cis-*EETs and total erythrocyte membrane *cis-*EETs in a Tr mouse model suggests that erythrocyte *cis-*EETs may be used as predictive markers for cardiac events. All *cis-*EET regioisomers displayed similar trends following acute MI; however, the magnitude of change for each regioisomer was markedly different, warranting measurement of each individually.

## 1. Introduction

Arachidonic acid (AA) is an ω-6 polyunsaturated fatty acid usually found esterified at the *sn-*2 position in phospholipid membranes. Upon activation of phospholipase A_2_ (PLA_2_), AA is released and metabolized through the cyclooxygenase, lipoxygenase, and CYP pathways [1]. Through the CYP pathway, primarily by CYP2C and CYP2J subfamilies, AA is metabolized to four bioactive *cis-*EETs [2]. These EETs can be rapidly hydrolyzed to dihydroxyeicosatrienoic acids (DHETs) by soluble epoxide hydrolase (sEH) [3]. EETs can be re-esterified into the phospholipid membrane via an acyl-coenzyme A-dependent mechanism or circulate in erythrocyte membranes, plasma lipoproteins, and fatty acid binding proteins [4,5,6,7].

EETs are established signaling molecules with numerous physiological functions, including anti-inflammatory, pro-angiogenesis, vasodilatory effects, and activation of several ion channels [8,9,10,11,12]. Extensive literature from animal studies and emerging evidence in humans suggest protective roles of EETs in cardiovascular events, especially during myocardial ischemia reperfusion injury.

Evidence that heart disease can alter *cis-*EET levels was suggested in patients with coronary artery disease (CAD). Higher plasma EETs were observed in CAD patients compared to healthy volunteers, although, obesity in CAD patients was associated with lower plasma EETs [13,14]. A follow up study from the same group showed total EET levels in both non-obstructive and obstructive CAD patients were generally lower than their control group (no apparent CAD) [15]. Due to the functional importance of EETs in cardiovascular health, monitoring levels of circulating EETs could conceivably serve as a reporter for cardiac events and a predictor of risk prior to the occurrence of the cardiovascular disease or injury. Performing heart biopsies to determine cardiac EET levels is not a viable option, thus alternatives are desirable, such as measuring circulating EETs. The percent distribution for each individual EET regioisomer relative to total EETs measured in rat cardiac tissue versus plasma, is similar in both compartments [7,16] supporting the concept that plasma EETs may be correlated with cardiac EETs. Another study in humans showed a strong correlation (r = 95%) between plasma and erythrocyte membrane levels of EETs [17]. Collectively, these published data suggest that EETs in erythrocyte membrane, or plasma, can potentially be used to report on EET levels in heart tissue, especially prior to a cardiac event, although there are no current studies to validate this correlation between erythrocyte membrane and cardiac EETs.

Cardiomyocyte-specific cytochrome P450 2J2 (CYP2J2) is the only CYP2J subfamily member expressed in humans [18]. Seven mouse *Cyp2j* genes and three *Cyp2j* pseudogenes have been identified to date. Most mouse *Cyp2j* isoforms are expressed extrahepatically. *Cyp2j6* transcript was found in high levels in small intestine and to a much lesser extent in the heart, while its unstable protein *in vitro* caused inconclusive western blot results [19]. *Cyp2j11* transcript is present in the kidney and heart, while the protein is expressed prominently in proximal convoluted tubules of kidney, liver, and to a lesser extent in cardiomyocytes [20]. Based on a comparison between AA and linoleic acid turnover rates, Cyp2j11 appeared to slightly prefer linoleic acid to AA and performed slightly more hydroxylation reactions than epoxygenation [20]. However, the authors did not account for non-enzymatic formation of EETs and the monohydroxylated metabolites of AA.

In this study, we assess the potential use of erythrocyte membrane EETs as surrogate markers for cardiac events in a Tr mouse model with cardiomyocyte-specific overexpression of human CYP2J2. The CYP2J2 Tr mice are more suitable to model human cardiac and erythrocyte membrane EET levels, given that the WT mice express several Cyp2j isoforms that are not all epoxygenases. Due to difficulties in obtaining matched cardiac and blood samples from the same subject, the transgenic mouse serves as a suitable alternative model. The effect of acute MI on both *cis-* and *trans-*EETs in cardiac tissue and erythrocyte membranes in the CYP2J2 Tr mice was also determined. In the model of acute MI, there is clear evidence of cardioprotection in the Tr mice compared to WT. Furthermore, CYP2J2 overexpression is associated with increased EET levels in both cardiac tissue and erythrocyte membranes.

## 2. Materials and Methods

### 2.1. Reagents and Solvents

Unless otherwise specified, all solvents used were purchased from Fisher Scientific (Waltham, MA, USA). Chemical reagents, PLA_2_, and thiazolyl blue tetrazolium bromide (MTT) were purchased from Sigma-Aldrich (St Louis, MO, USA). All deuterated and non-deuterated *cis*-EETs used for internal standards, standard curves, and quality controls were purchased and obtained from Cayman Chemical (Ann Arbor, MI, USA). Adult human derived ventricular myocytes were obtained from Celprogen, Inc. (Torrance, CA, USA). CYP2J2 human small interfering RNA (siRNA) Trilencer oligo duplexes were obtained from Origene(Rockville, MD, USA). RNAi MAX lipofectamine and OptiMEM reduced serum media were obtained from Thermo Fisher Scientific (Waltham, MA, USA).

### 2.2. Standard Curve

For both erythrocyte membranes and cardiac tissue samples, final concentrations for each regioisomer of *cis-*EET were 0, 5, 10, 25, 50, 75, 100, 250, and 500 ng/mL to make up the standard curve. Two quality control samples were used at 10 and 100 ng/mL of *cis-*EETs. Quality controls were diluted and prepared from original stocks separately from calibration curve standards. Calibration curve samples were treated and extracted in the same manner as the biological samples. The only exception was during the PLA_2_ hydrolysis step; the dried calibration curve extracts were dissolved in 1× PLA_2_ buffer followed by addition of nitrogen-purged ethyl acetate instead of the enzyme. The rest of the extraction procedure followed the same steps previously described [21].

### 2.3. Animal Model

Mice with cardiomyocyte-specific CYP2J2 overexpression driven by the α-myosin heavy chain promoter (α-MHC) were described previously [22]. Tr male mice were obtained from Dr. Darryl Zeldin’s lab at National Institute of Environmental Sciences. C57BL/6 WT female mice were purchased from Charles River Laboratories (Wilmington, MA, USA) and bred with Tr male mice to produce a heterozygote offspring expressing CYP2J2. Mice were bred in-house and managed by the University of Washington, Department of Comparative Medicine. Pups were weaned at 21 days, at which time a tail biopsy was obtained. Genotyping and assignment of a Tr or WT genotype was performed following procedures previously published [23]. All animal experiments were performed in compliance with approved protocols by Institutional Animal Care and Use Committee of the University of Washington.

Mice of both genotype and gender (4 to 23 months old) were used for studies correlating EET levels in heart vs. erythrocyte membrane. To study the effect of MI on EET levels, mice were separated into 4 groups. In each group, 4 male and 4 female mice 3–4 months old were used. The groups were as follows: WT sham, Tr sham, WT-MI, and Tr-MI.

### 2.4. Echocardiography and MI Surgery

Mice were anesthetized with 1.5% isoflurane prior to echocardiography. Echocardiographic imaging was performed with a VisualSonics Vevo 2100 imaging system (Fujifilm VisualSonics, Inc., Toronto, ON, Canada) as described previously [24]. Briefly, M-mode ventricular dimensions were measured and averaged from 3 to 5 cycles. Fractional shortening (FS) was calculated using the following equation:$$FS=\left[\frac{\left(LVED-LVES\right)}{LVED}\right]\times 100\%.$$

*LVES* is left ventricular dimension at the end of systole, whereas, *LVED* is dimension of left ventricle at the end of diastole. Echocardiography was performed prior to surgery, a week post-surgery, and two weeks post-surgery prior to tissue harvesting. The surgical procedure for MI in the mouse with permanent ligation of the left anterior descending artery has been described previously [25]. A similar procedure without ligation was performed on sham mice.

### 2.5. Histological Analysis, Cell Surface Area Measurement, and Terminal Deoxynucleotidyl Transferase dUTP Nick End Labelling (TUNEL)

Histological analysis of fibrosis, and cell surface area measurements were described previously [25]. Briefly, mouse hearts (N = 4–5) were fixed in 10% formalin/phosphate-buffered saline (PBS) and dehydrated for paraffin embedding. The dehydrated, paraffin-embedded fixed hearts were then cut into 5 μm sections for Masson’s trichrome staining. The percentage of myocardial fibrosis was determined using the ratio of the total interstitial fibrosis area to longitudinal sectional area of a left ventricle section using MetaMorph 6.1 software (Molecular Devices, LLC., San Jose, CA, USA) as described previously [25]. Using a section of dehydrated, paraffin-embedded fixed hearts, TUNEL was performed using TMR Red in situ cell death detection kit following the manufacturer’s instructions.

### 2.6. Detection of ROS

Using frozen cardiac sections, cellular production of ROS was detected using chloro methyl- 2’,7’-dichlorodihydrofluorescein diacetate (CM-H_2_DCFDA) staining following the manufacturer’s instruction. The percentage of ROS generated was calculated as ratio of the total fluorescent area to the whole area using ImageJ software (National Institutes of Health, Bethesda, MD, USA). The values reported for % ROS were averages from 4~5 images.

### 2.7. Extraction of EETs from Erythrocyte Membrane and Heart Tissue

Extraction of EETs from both erythrocyte membranes and heart tissue followed a procedure detailed previously in Aliwarga et al. [21]. The only exception is that EETs extracted from both erythrocyte membranes and cardiac tissue were reconstituted in 50 µL solution containing 50% water and 50% of 80:20 acetonitrile:methanol prior to mass spectrometric (MS) analysis.

### 2.8. Liquid Chromatography and MS Assay to Quantify EETs

Quantification of *cis*- and *trans*-EETs was performed as previously described [21].

### 2.9. Human Cardiomyocytes and Cobalt (II) Chloride (CoCl_2_) Treatment

The role of CYP2J2 in cellular hypoxic response was determined using a commercially available adult human-derived ventricular myocytes. All experiments were performed within eight passages from receipt of the cells from Celprogen (Torrance, CA, USA). The cells were cultured and passaged per Celprogen’s instructions. In these experiments, cells were plated onto 12-well plates at a density of 50k per well. *CYP2J2* gene expression was decreased by at least 80% using CYP2J2 human siRNA Trilencer oligo duplexes delivered by RNAi MAX lipofectamine using protocols previously described [26]. Scrambled siRNA was similarly prepared in lipofectamine and used to treat control cells.

Briefly, the siRNA was prepared with the lipofectamine using OptiMEM reduced serum media to a concentration of 50 nM. Cells were washed with warm (37 °C) PBS and harvested using trypsin. Following trypsinization, cells were pelleted, resuspended, and diluted to a concentration of 100,000 cells/mL in complete medium. The lipofectamine/siRNA stocks were added to each well to a final concentration of 10 nM siRNA, followed by addition of the cells. Cells were incubated with the lipofectamine/siRNA for 72 h, after which cells were exposed to CoCl_2_ (20, 40 and 60 μM) for 24 h.

After 24 h of CoCl_2_ exposure, MTT assays were performed to determine the relative cell viability among the treatment groups as previously described [26]. Cells were exposed to MTT (60 μM) and incubated for 20 min at 37 °C. The medium was then carefully removed, and the colorimetric dye was resuspended in dimethyl sulfoxide (DMSO) (600 μL) and Sorenson’s glycine (75 μL, 100 mM glycine and 100 mM sodium chloride). The absorbance in each well was measured using a Tecan Infinite M200 plate reader (Tecan, Männedorf, Switzerland) using the following protocol: Five seconds of orbital shaking with an amplitude of 1 mm, followed by 10 s of wait time, and then absorbance measurement at 570 nm (9 nm bandwidth) using 670 nm (9 nm bandwidth) as the reference wavelength. A true zero signal was obtained by following the same protocol using well with no cells. Measurements were normalized to the absorbance in CoCl_2_ untreated control wells (set as 100% viability).

The effects of EET pretreatment on CoCl_2_ toxicity were assessed with EET rescue experiments. *CYP2J2* gene expression was silenced with siRNA as described above; however, prior to CoCl_2_ treatment, cells were exposed to a mixture of all four regioisomers of EETs (50 nM final concentration) for 60 min. After 60 min the cells were treated with CoCl_2_ for 24 h as described above followed by an MTT assay. Cells not treated with CoCl_2_ were used for normalization.

### 2.10. Data Analysis

Mass spectrometry data were analyzed using MassLynx 4.1 (Waters Corporation, Milford, MA, USA). Experimental data analysis was performed using Prism 7.04 (GraphPad, La Jolla, CA, USA). Correlation analyses were performed using non-parametric Spearman’s correlation as denoted by r_s_ values.

In the biological samples, peak height of each analyte was normalized to the corresponding peak height of internal standard. Levels of each *cis-*EET regioisomer were quantified using a standard curve, while levels of *trans-*EETs were determined using normalized peak height ratios (PHR). In addition to analyte normalization to the corresponding internal standard, the amount or ratio of peak heights for each analyte were normalized to total protein content for erythrocyte membrane and cardiac tissue. Since only one fifth of the heart homogenate was used for each sample, the amount or ratios of peak heights of each analyte were multiplied by five to scale for the entire heart.

## 3. Results

### 3.1. EET Levels and Correlation in Erythrocyte Membrane and Cardiac Tissue from WT and Tr Mice

No difference was observed in total EET levels esterified in erythrocyte membranes of WT and Tr mice (Figure 1A). However, a significant increase in total *cis-* and *trans*-EETs from cardiac tissue was observed in Tr mice compared to WT mice (Figure 1B). In addition, there was no difference observed in total EET levels in both erythrocyte membranes and cardiac tissues between male and female mice (data not shown).

The correlation between EET levels in erythrocyte membrane and cardiac tissue was also examined. In WT mice, no correlation was observed between EET levels in erythrocyte membranes and cardiac tissue, except for *cis-*5,6-EET (r_s_ = 0.4769, *p* = 0.0159; Figure 2 and Appendix A). In Tr mice, both 14,15- (r_s_ = 0.6215, *p* = 0.0009) and 11,12-EET (r_s_ = 0.7062, *p* < 0.0001) as well as total EET levels (r_s_ = 0.4392, *p* = 0.0280) in erythrocyte membranes and cardiac tissue were significant and positively correlated (Figure 3 and Appendix A).

### 3.2. Cis- and Trans-EET Levels, Cardiac Morphology, and Function Following MI

WT and Tr mice were subjected to MI or sham surgery for two weeks. In WT mice, *cis-*EET levels from erythrocyte membranes were not significantly affected by MI. While, in Tr mice, MI significantly increased *cis-*EET levels in erythrocyte membrane with the exception of *cis-*5,6-EET (Figure 4). Intriguingly, while MI significantly increased the levels of all regioisomers of *trans-*EETs in Tr mice, MI significantly decreased *trans-*5,6-EET and total *trans-*EET levels from erythrocyte membrane in WT mice (Appendix A). Total *cis-*EETs in erythrocyte membranes were higher than total *trans-*EETs in all treatment groups (Appendix A).

In cardiac tissue, *cis-* and *trans-*EETs in both WT and Tr mice were significantly increased following MI (Figure 5 and Appendix A). However, in contrast to what was observed in erythrocyte membrane, total *trans-*EETs in cardiac tissue were higher than total *cis-*EETs (Appendix A). In WT mice, MI significantly increased *trans-*8,9- and *trans-*11,12-EETs in cardiac tissue. In Tr mice, EET levels were significantly increased in both erythrocyte membrane and cardiac tissue.

To investigate the pathophysiologic effects of cardiac-specific CYP2J2 overexpression, which are associated with higher EET levels in cardiac tissue, echocardiographic and histological analyses were performed in Tr vs. WT mice subjected to MI. Cardiac contractility was reduced in both groups as represented by lower percent FS; however, Tr mice retained significantly better cardiac contractility compared to WT mice (Figure 6A). The ratio of lung weight to body weight was significantly increased in WT mice following MI, an indicator of pulmonary edema associated with congestive heart failure (Figure 6B,C). This effect was significantly ameliorated in Tr mice. Histological analysis indicate that cardiomyocyte-specific overexpressing CYP2J2 Tr mice were protected from pathological remodeling associated with MI injury while displaying significantly smaller infarct size and lower myocardial fibrosis compared to their WT counterparts (Figure 7A–D).

The extent of apoptosis due to MI was determined using a TUNEL assay. Tr mice subjected to MI had fewer apoptotic cells compared to their WT counterparts (Figure 7E,F). Furthermore, myocardial ROS production was significantly reduced in Tr-MI mice compared with WT-MI mice (Figure 7G).

### 3.3. Effect of Hypoxia on Cardiac Cells in Human Cardiomyocytes with Altered CYP2J2 Expression

The role of CYP2J2 in protecting against hypoxia-induced toxicity in human cardiomyocytes exposed to CoCl_2_ (Figure 8) was investigated. CYP2J2 gene knockdown using siRNA decreased cell viability following CoCl_2_ treatment compared to cells with baseline *CYP2J2*. At 20 and 40 μM CoCl_2_, cells treated with CYP2J2 siRNA were significantly less viable compared to cells treated with scrambled siRNA. At 60 μM CoCl_2_, there was no difference in cell viability between the two groups. Pretreatment with a mixture of the four regioisomers of EETs (50 nM total), increased the viability of cells exposed to CoCl_2_ (Figure 8). At 20 μM CoCl_2_, EET pretreatment increased cell viability from ~40% to about 50–60%. At higher CoCl_2_ concentration, cell viability was increased from ~20% to 40–70% at 40 μM and from ~10% to 30–50% at 60 μM.

## 4. Discussion

A major goal of this study was to determine if EETs are altered in cardiac tissue during acute injury and if this change can be captured in circulating levels. Cardiomyocyte-specific overexpressing CYP2J2 Tr mice and WT controls were used to measure EETs in matched cardiac tissue and erythrocyte membranes. Since Tr mice express human CYP2J2, EET formation and regioisomer distribution should mimic human cardiac tissue and serve as a better model than WT mice. Tr mice exhibited significantly higher *cis-* and *trans-*EETs in cardiac tissue compared to WT, while EET levels in erythrocyte membranes were similar (Figure 1). No correlation was observed in *trans-*EETs extracted from erythrocyte membrane and cardiac tissue of WT and Tr mice, with the exception of *trans-*14,15- and *trans-*11,12-EETs in Tr mice. *Cis-*EETs in erythrocyte membrane and cardiac tissue were not correlated in WT mice. However, a significant and positive correlation was observed between erythrocyte membrane and cardiac tissue *cis-*EETs extracted from Tr mice. These findings suggest that *cis-*EETs, not *trans*-EETs, which are not a product of an enzymatic pathway, in erythrocyte membrane could potentially report on cardiac *cis*-EET levels. To validate this finding, it would be ideal to administer a specific CYP2J2 inhibitor to WT and transgenic mice and measure cardiac tissue and erythrocyte membrane EETs. Unfortunately, to date, a CYP2J2 specific inhibitor useful for in vivo studies is not available.

To elucidate the cardioprotective effects of EETs and determine if they are altered due to cardiac injury, we subjected Tr and WT mice to acute MI for two weeks. There have been two conflicting reports on the effect of CAD on circulating EET levels. Theken et al. reported that CAD patients had higher plasma EET levels than healthy volunteers, while a follow up study reported that total plasma EET levels in obstructive CAD patients were significantly lower than the control group (no apparent CAD) [13,15]. CAD patients in those earlier studies had experienced a chronic disease progression, while the mice in this study were exposed to acute MI. The species differences and dissimilar exposure to the disease made it challenging to predict how EET levels would be altered *a priori*. We found that, in response to MI-induced cardiac stress, levels of most regioisomers of *cis-*EETs and total *cis-*EETs were significantly higher in both erythrocyte membranes and cardiac tissue of Tr mice compared to their sham counterparts (Figure 4 and Figure 5). Similarly, *trans-*EETs in both erythrocyte membranes and cardiac tissue of Tr mice also increased following MI (Appendix A). Our observations indicate EETs increase in response to acute MI with concomitant protection against ischemia-associated cardiac events. In particular, echocardiographic data demonstrated cardiac contractility of Tr-MI, indicated by % FS, that was significantly better than WT-MI (Figure 6A). Tr-MI also had significantly lower pulmonary edema compared to WT-MI (Figure 6C). Furthermore, Tr-MI mice exhibited significantly lower infarction and fibrosis area with significantly fewer apoptotic cells and reduced ROS production (Figure 7). These results provide strong evidence that cardiac-specific overexpression of CYP2J2 is protective against the sequelae of acute MI. All data collected from this mouse model subjected to MI, support the hypothesis that higher EET levels confer protection against acute MI. These results agree with several studies demonstrating the protective nature of overexpressing CYP2J2 in mouse cardiac tissue [22,27,28,29], but for the first time, also measures changes in cardiac tissue and erythrocyte membrane EETs following MI. 

To corroborate these findings in a human-derived model system, we showed that *CYP2J2* expression also has a protective effect using adult-derived cardiomyocytes, (Figure 8). Consistent with the mouse data, adult ventricular myocytes were protected from CoCl_2_-induced hypoxia when expressing basal levels of *CYP2J2*. But when levels were decreased using siRNA, cells became more sensitive to the toxic effects of CoCl_2_. Conversely, when cells were exposed to EETs (50 nM) prior to CoCl_2_ treatment, they were protected from CoCl_2_ cytotoxicity. It is important to note that the CoCl_2_ concentrations used in these experiments do not inhibit CYP2J2 activity (Appendix A).

Two puzzling results remain unexplained from this study. The first is the higher levels of cardiac *trans-*EETs in both WT and Tr mice compared to their *cis-*EETs counterparts (Appendix A). *Trans-*EETs formation in vivo is likely due to free radical oxidation processes of arachidonic acid esterified in the membrane [21]. In healthy heart tissue, ROS are always present as byproducts from mitochondrial electron transport chain. Mitochondria constitute about 20–40% of cardiac tissue cellular volume [30]. In addition, production of ROS is increased in non-infarcted left ventricular myocardium following MI in CD-1 mouse model [31]. As shown in Figure 6C,D, cardiac tissue of Tr-MI mouse exhibited myocardial fibrosis although lower than WT-MI. A potential explanation is that the excess mitochondrial ROS production due to MI led to formation of more *trans-*EETs than *cis-*EETs in cardiac tissue of MI mice.

The second unexpected finding was the higher levels of cardiac *trans-*EETs in Tr-MI mice compared to WT-MI mice, even though ROS levels in Tr-MI were significantly lower than in WT-MI. The magnitude of EETs increase due to MI in WT and Tr mice were similar reflected by the observation that *trans-* to *cis-* ratios of each regioisomer of EET extracted from cardiac tissue did not change significantly (Table 1). Levels of cardiac *trans-*EETs of untreated Tr mice at baseline were significantly higher compared to that of WT mice (Figure 1B). Basal contractile function and cardiac anatomy of the Tr mice were found to be normal [22]. However, the impact of inserting and overexpressing CYP2J2 transgene on the expression and regulation of other enzymes and pathways remains unknown. Chaudhary et al. reported no change in expression or activity of cardiac sEH in WT C57BL/6 and Tr mice [32]. However, the ability of sEH to hydrolyze *trans-*EETs in WT vs. Tr mice remains unknown. Perhaps the clearance pathways of *cis*- vs. *trans-*EETs are altered in disease state.

A couple of available cardiac therapies are dietary supplementation of α-lipoic acid (ALA) and cardiac resynchronization therapy (CRT). ALA, an enzymatic cofactor involved in energy production and mitochondrial biogenesis, has antioxidant and anti-inflammatory properties [33]. Dietary supplementation of ALA in two mouse models for human atherosclerosis exhibited significant reduction in atherosclerotic lesion formation [34]. In patients with paroxysmal, symptomatic atrial fibrillation who underwent catheter ablation, dietary supplementation of ALA reduced the serum levels of inflammation markers [35]. In a lipopolysaccharide -challenged C57BL/6J mouse model, administration of ALA attenuated cardiac dysfunction and myocardial inflammatory response by preserving the PI3K/Akt pathway [36]. Pretreatment of EETs prior to hypoxia/reoxygenation in primary neonatal rat cardiomyocytes exhibited improved survival and attenuated apoptosis by activating several pathways including the PI3K/Akt pathway [37]. It is possible that supplementation of ALA mitigates inflammation associated with cardiac events, while EETs protect the cardiomyocytes through activation of the PI3K/Akt pathway.

In ischemic or non-ischemic cardiomyopathic patients with QRS duration of 150 msec or greater, CRT was associated with better prognosis by significantly reducing the volume of left ventricle and improving ejection fraction [38]. The cardiac remodeling processes associated with CRT are related to alterations in genes and miRNA expression [39,40]. Reports on how miRNA affects CYP2J2 during cardiac events are currently nonexistent. miRNA-let-7b has been suggested to decrease the expression of CYP2J2 in several cancer cell lines [41] but no data is available in cardiac cells. Expression of CYP2J2 in U87 cells, a human glioblastoma cell line, significantly decreased when miRNA-584 was co-transfected in the cells [42]. Inhibition of miRNA-let7, specifically miRNA-let-7c, was shown to prevent post-MI remodeling and to improve cardiac function [43]. However, when Marfella et al. compared miRNA-let-7c expression between CRT-responsive and CRT-non-responsive patients, there was no significant difference between the two groups of patients [40].

In summary, Tr mice had improved overall cardiac outcome following acute MI, compared to WT mice, likely due to increased production of cardiac EETs. Elevated cardiac *cis*-EETs in Tr mice following MI were also reflected by circulating levels in erythrocyte membranes. The translation of these results to humans and determining if circulating *cis-*EETs can report on chronic or incident cardiac events warrant future studies.

In past decades, several new CVD diagnostic and prognostic biomarkers that are expressed locally at the site of inflammation and circulate in the peripheral blood have been identified [44]. In this study, we reported that *cis-*EETs could potentially be used as a marker to report CVD events, although a validation study using a specific CYP2J2 inhibitor is still needed. We show that acute MI elevates EET levels to potentially help reduce the pathophysiological consequences on cardiac tissue. Incorporating a CYP2J2 transgene in cardiomyocyte of a mouse model demonstrated profound protective effects as a result of having higher EET levels, supported by echocardiographic, cellular, and histological data. Lastly, given that the levels of each regioisomer of EET change independently with disease severity, it will be useful to investigate the function of each regioisomer of EETs and not just total EETs. Of note, alterations in *cis-*EET levels are mimicked by the *trans-*EETs. Therefore, the function, regulation, and clearance of *trans-*EETs in vivo in the presence of stress needs to be further explored.

## Figures and Tables

**Figure 1 biomedicines-08-00144-f001:**
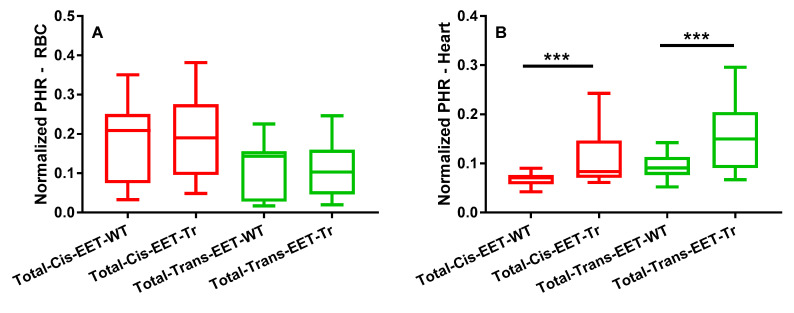
Total *cis-* and *trans-*epoxyeicosatrienoic acids (EETs) extracted from (**A**) erythrocyte membrane and (**B**) cardiac tissue of wild-type (WT) mice (N = 25, 18 female and seven male, *** *p* ≤ 0.001) and cardiomyocyte-specific overexpressing Cytochrome P450 2J2 (CYP2J2) transgenic (Tr) mice (N = 25, 16 female and nine male, *** *p* ≤ 0.001).

**Figure 2 biomedicines-08-00144-f002:**
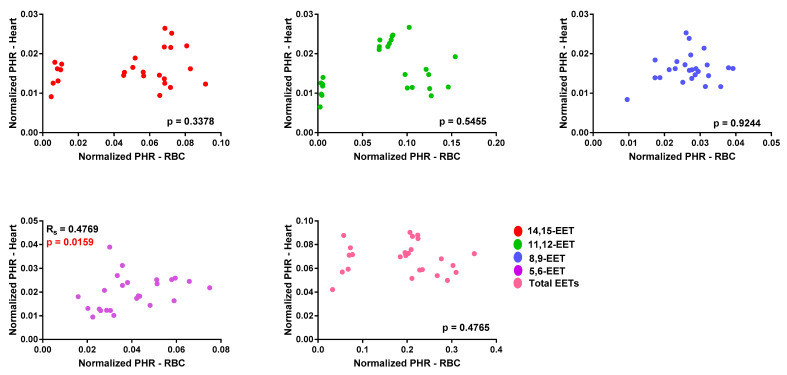
Correlation of each *cis-*epoxyeicosatrienoic acid (EET) regioisomer and total *cis-*EETs in erythrocyte membrane and cardiac tissue of wild-type mice. Relative amount of each regioisomer of EETs was represented by normalized peak height ratio peak height ratio (PHR) in erythrocyte membrane or cardiac tissue.

**Figure 3 biomedicines-08-00144-f003:**
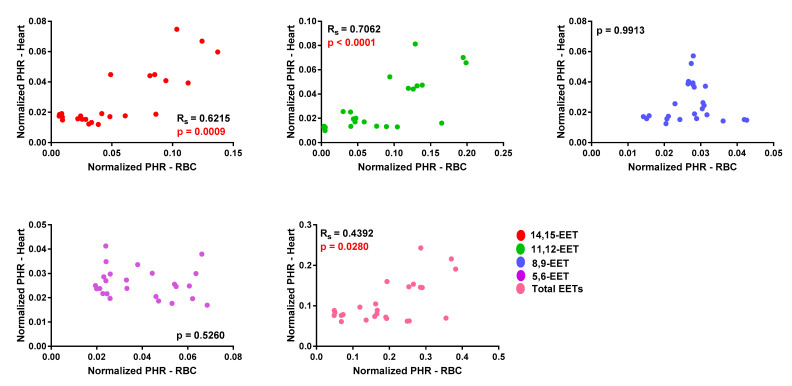
Correlation of each regioisomer of *cis-*epoxyeicosatrienoic acid (EET) and total *cis-*EETs in erythrocyte membrane and cardiac tissue of CYP2J2 transgenic mice. Relative amount of each EET regioisomer was represented by normalized peak height ratio peak height ratio (PHR) in erythrocyte membrane and cardiac tissue.

**Figure 4 biomedicines-08-00144-f004:**
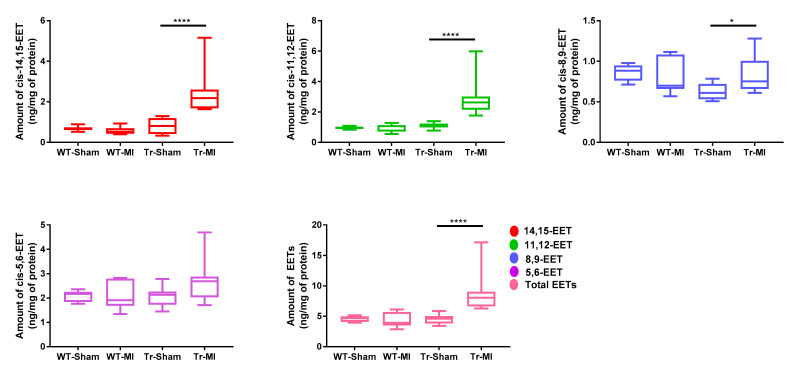
Levels of *cis-*epoxyeicosatrienoic acid (EET) regioisomers and total *cis-*EETs extracted from erythrocyte membrane of wild-type (WT) and transgenic (Tr) mice subjected to sham or myocardial infarction (MI) surgery. Eight WT mice (4 female (F) and 4male (M) mice) and nine Tr mice (4 F and 4 M were subjected to sham surgery, and 9 WT mice (4 F and 5 M mice) and nine Tr mice (4 F and 5 M mice) were subjected to MI surgery. * indicates *p* ≤ 0.05 and **** indicates *p* ≤ 0.0001.

**Figure 5 biomedicines-08-00144-f005:**
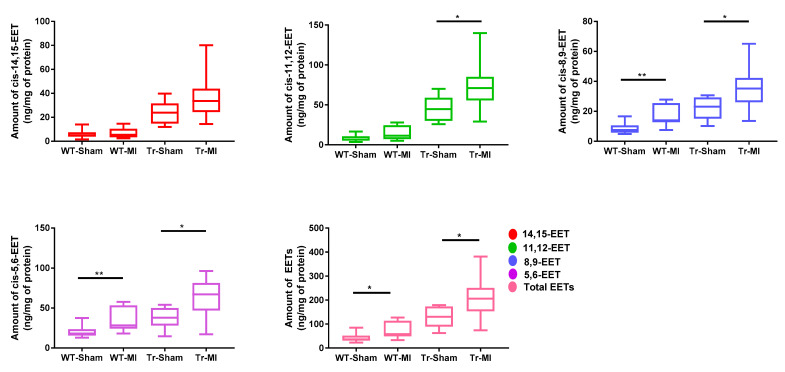
*Cis-*epoxyeicosatrienoic acid (EET) regioisomers and total *cis-*EET levels extracted from cardiac tissue of wild-type (WT) and transgenic (Tr) mice subjected to sham or myocardial infarction (MI) surgery. Eight WT mice (four female (F) and four male (M)) and nine Tr mice (five F and four M) were subjected to sham surgery and nine WT mice (four F and five M) and nine Tr mice (four F and five M) were subjected to MI surgery. * indicates *p* ≤ 0.05 and ** indicates *p* ≤ 0.01.

**Figure 6 biomedicines-08-00144-f006:**
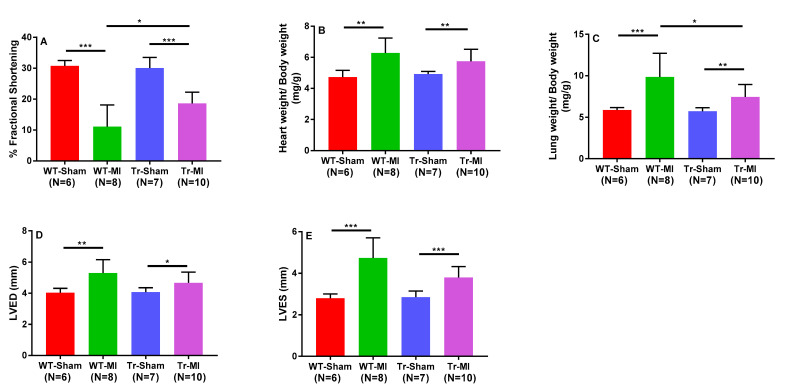
Echocardiographic measurement of ventricular function following myocardial infarction (MI). (**A**) Percentage fractional shortening (%FS), an indicator of contractile function of the left ventricle; (**B**) heart weight to body weight (HW/BW) ratio; (**C**) lung weight to body weight (LW/BW) ratio; (**D**,**E**) measurement of the left ventricle dimensions at the end of diastole (the biggest cardiac dimension, LVED, D) and systole (the smallest cardiac dimension, LVES, E) in mice subjected to sham or MI surgery. * Indicates *p* < 0.05, ** *p* < 0.005, *** *p* < 0.0005.

**Figure 7 biomedicines-08-00144-f007:**
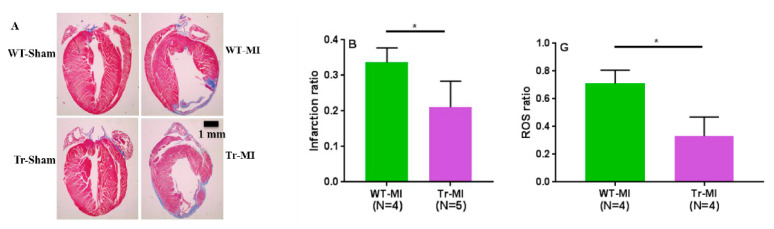
Paraffin-embedded cardiac sections of WT and Tr mice were stained using Masson’s Trichrome (**A**,**C**). (**A**) Shows whole slice of hearts, while (**C**) shows only infarct zones. Blue area indicates myocardial fibrosis, while red area indicates healthy muscle tissue. (**B**) Myocardial infarction size ratio and (**D**) myocardial fibrosis ratio were determined using MetaMorph 6.1 software. (**E**) Terminal Deoxynucleotidyl Transferase dUTP Nick End Labelling (TUNEL)-positive nuclei of paraffin-embedded sections from the mice hearts indicated in (**A**). Apoptotic cells appear brown, while normal cells appear blue. (**F**) Measurement of TUNEL-positive nuclei in cardiac sections from mice indicated in (**A**). (**G**) Measurement of reactive oxygen species (ROS) ratio using chloro methyl- 2’,7’-dichlorodihydrofluorescein diacetate (CM-H_2_DCFDA) staining from mice frozen cardiac sections. * Indicates *p* ≤ 0.05.

**Figure 8 biomedicines-08-00144-f008:**
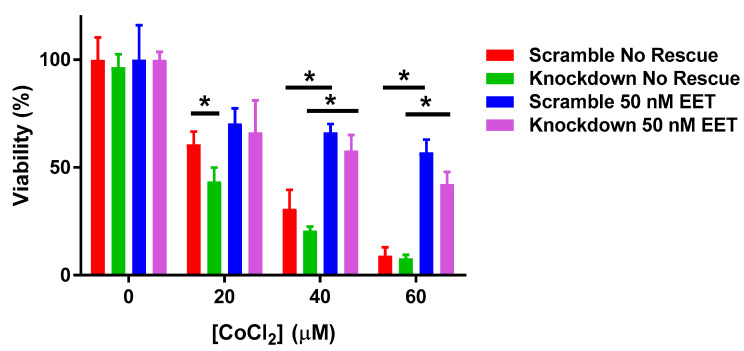
Cell viability in human adult-derived ventricular myocytes exposed to varying doses of CoCl_2_ for 24 h, following 72 h of lipofectamine treatment with either scrambled siRNA or CYP2J2 siRNA and addition of 50 nM total EETs prior to CoCl_2_ treatment. * Indicates *p* < 0.05.

**Table 1 biomedicines-08-00144-t001:** Summary of geometric isomers of individual regioisomer of EETs in cardiac tissue and erythrocyte membrane of WT and Tr mice subjected to sham or MI surgery. There were eight WT mice (4 female and 4 male) and nine Tr mice (5 female and 4 male) subjected to sham surgery and nine WT mice (4 female and 5 male) and nine Tr mice (4 female and 5 male) subjected to MI surgery.

	Heart					Erythrocyte				
	*cis-/trans-* ratio					*cis-/trans-* ratio				
	14,15-EET	11,12-EET	8,9-EET	5,6-EET	Total	14,15-EET	11,12-EET	8,9-EET	5,6-EET	Total
WT-Sham	0.59 ± 0.1	0.66 ± 0.1	0.47 ± 0.1	0.18 ± 0.08	0.30 ± 0.06	2.2 ± 0.3	1.9 ± 0.2	2.4 ± 0.4	1.4 ± 0.2	1.8 ± 0.3
Tr-Sham	0.53 ± 0.05	0.78 ± 0.09	0.27 ± 0.04	0.11 ± 0.005	0.28 ± 0.02	2.2 ± 0.4	1.9 ± 0.2	1.8 ± 0.6	1.6 ± 0.3	1.8 ± 0.1
WT-MI	0.50 ± 0.1	0.60 ± 0.1	0.56 ± 0.1	0.38 ± 0.09	0.47 ± 0.1	2.0 ± 0.2	1.8 ± 0.2	2.6 ± 0.3	1.9 ± 0.2	2.0 ± 0.2
Tr-MI	0.49 ± 0.07	0.68 ± 0.1	0.25 ± 0.04	0.13 ± 0.01	0.28 ± 0.03	2.4 ± 0.2	1.8 ± 0.3	1.3 ± 0.4	1.5 ± 0.3	1.7 ± 0.2

Each entry is an average of eight to nine ratios of geometric isomers of EETs. Data are presented as mean ± S.D.

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
