# Peer review of "Higher Epoxyeicosatrienoic Acids in Cardiomyocytes-Specific CYP2J2 Transgenic Mice Are Associated with Improved Myocardial Remodeling"

_biomedicines, 2020, doi:10.3390/biomedicines8060144_

Round 1
Reviewer 1 Report
In this study, Aliwarga and colleagues address an interesting clinical topic: Association of CYP2J2 and EET levels to cardiovascular disease. The authors elegantly demonstrated the cardioprotective effects of EETs in cardiomyocytes specific CYP2J2 transgenic mice. Moreover, the authors showed that erythrocyte cis-EETs could be a potential marker for cardiovascular disease. Overall, the scientific design of the study is sound, the complementarity of the experimental approaches is remarkable, and the take-home message is clear. However, wondering, did authors check the expression of soluble epoxide hydrolase in their transgenic mice.
Author Response
We would like to thank Reviewer 1 for evaluating our manuscript and positive feedback offered. The expression of soluble epoxide hydrolase (sEH) in our transgenic mice has been elucidated by Chaudhary et al. They found that there was no difference in cardiac sEH expression in both C57BL/6 wild-type and cardiomyocyte-specific overexpressing CYP2J2 transgenic mice. Chaudhary et al. also showed hydrolysis of cis-14,15-EET to form cis-14,15-DHET remain unaffected in transgenic mice compared to their wild-type counterparts. We have cited this paper now and added the findings by Chaudhary et al. in the discussion section of the manuscript lines 373-377.
Reviewer 2 Report
Excellent research on experimental mice model about potential protective role of EET during myocardial infarction in relation to cardiac remodeling severity and manifestations of heart failure. I have no remarks and corrections. Thanks a lot!
Author Response
We would like to thank Reviewer 2 for evaluating our manuscript and positive feedback offered.
Comments and suggestions for authors: Excellent research on experimental mice model about potential protective role of EET during myocardial infarction in relation to cardiac remodeling severity and manifestations of heart failure. I have no remarks and corrections. Thanks a lot!
Response: We would like to thank Reviewer 2 for evaluating our manuscript and positive feedback offered.